# Learning Chordal Markov Networks by Constraint Satisfaction

**Jukka Corander**[*†]
University of Helsinki
Finland

**Tomi Janhunen**[*‡]
Aalto University
Finland

**Jussi Rintanen**[*‡§]
Aalto University
Finland

**Henrik Nyman**[¶]
Åbo Akademi University
Finland

**Johan Pensar**[¶]
Åbo Akademi University
Finland

## Abstract

We investigate the problem of learning the structure of a Markov network from data. It is shown that the structure of such networks can be described in terms of constraints which enables the use of existing solver technology with optimization capabilities to compute optimal networks starting from initial scores computed from the data. To achieve efficient encodings, we develop a novel characterization of Markov network structure using a balancing condition on the separators between cliques forming the network. The resulting translations into propositional satisfiability and its extensions such as maximum satisfiability, satisfiability modulo theories, and answer set programming, enable us to prove optimal certain networks which have been previously found by stochastic search.

## 1 Introduction

Graphical models (GMs) represent the backbone of the generic statistical toolbox for encoding dependence structures in multivariate distributions. Using Markov networks or Bayesian networks conditional independencies between variables can be readily communicated and used for various computational purposes. The development of the statistical theory of GMs is largely set by the seminal works of Darroch et al. [1] and Lauritzen and Wermuth [2]. Although various approaches have been developed to generalize the theory of graphical models to allow for modeling of more complex dependence structures, Markov networks and Bayesian networks are still widely used in applications ranging from genetic mapping of diseases to machine learning and expert systems.

Bayesian learning of undirected GMs, also known as *Markov random fields*, from databases has attained a considerable interest, both in the statistical and computer science literature [3, 4, 5, 6, 7, 8, 9]. The cardinality and complex topology of GM space pose difficulties with respect to both the computational complexity of the learning task and the reliability of reaching representative model structures. Solutions to these problems have been proposed in earlier work. Della Pietra et al. [10] present a greedy local search algorithm for Markov network learning and apply it to discovering word morphology. Lee et al. [11] reduce the learning problem to a convex optimization problem that is solved by gradient descent. Related methods have been investigated later [12, 13].

---

[*]This work was funded by the Academy of Finland, project 251170.

[†]Funded by ERC grant 239784.

[‡]Also affiliated with the Helsinki Institute of Information Technology, Finland.

[§]Also affiliated with Griffith University, Brisbane, Australia.

[¶]This work was funded by the Foundation of Åbo Akademi University, as part of the grant for the Center of Excellence in Optimization and Systems Engineering.

Certain types of stochastic search methods, such as Markov Chain Monte Carlo (MCMC) or simulated annealing, can be proven to be consistent with respect to the identification of a structure maximizing posterior probability [4, 5, 6, 7]. However, convergence of such methods towards the areas associated with high posterior probabilities may still be slow when the number of nodes increases [4, 6]. In addition, it is challenging to guarantee that the identified model indeed truly represents the global optimum since the consistency of MCMC estimates is by definition a limit result. To the best of our knowledge, strict constraint-based search methods have not been previously applied in learning of Markov random fields. In this article, we formalize the structure of Markov networks using constraints at a general level. This enables the development of reductions from the structure learning problem to *propositional satisfiability* (SAT) [14] and its generalizations such as *maximum satisfiability* (MAXSAT) [15], and *satisfiability modulo theories* (SMT) [16], as well as *answer-set programming* (ASP) [17]. A main novelty is the recognition of maximum weight spanning trees of the clique graph by a condition on the cardinalities of occurrences of variables in cliques and separators, which we call the *balancing condition*.

The article is structured as follows. We first review some details of Markov networks and the respective structure learning problem in Section 2. To enable efficient encodings of Markov network learning as a constraint satisfaction problem, in Section 3 we establish a new characterization of the separators of a Markov network based on a *balancing condition*. In Section 4, we provide a high-level description how the learning problem can be expressed using constraints and sketch the actual translations into propositional satisfiability (SAT) and its generalizations. We have implemented these translations and conducted experiments to study the performance of existing solver technology on structure learning problems in Section 5 using two widely used datasets [18]. Finally, some conclusions and possibilities for further research in this area are presented in Section 6.

## 2   Structure Learning for Markov Networks

An undirected graph $G = \langle V, E \rangle$ consists of a set of *nodes* $V$ which represents a set of random variables and a set of *undirected edges* $E \subseteq \{\{n, n'\} \mid n, n' \in V \text{ and } n \neq n'\}$. A *path* in a graph is a sequence of nodes such that every two consecutive nodes are connected by an edge. Two sets of nodes $A$ and $B$ are said to be *separated* by a third set of nodes $D$ if every path between a node in $A$ and a node in $B$ contains at least one node in $D$. An undirected graph is *chordal* if for all paths $n_0, \dots n_k$ with $k \geq 4$ and $n_0 = n_k$ there exist two nodes $n_i, n_j$ in the path connected by an edge such that $j \neq i \pm 1$. A *clique* in a graph is a set of nodes $c$ such that every two nodes in it are connected by an edge. In addition, there may not exist a set of nodes $c'$ such that $c \subset c'$ and every two nodes in $c'$ are connected by an edge. Given the set of cliques $C$ in a chordal graph, the set of *separators* $S$ can be obtained through intersections of the cliques ordered in terms of a junction tree [19], this operation is considered thoroughly in Section 3.

A Markov network is defined as a pair consisting of a graph $G$ and a joint distribution $P_V$ over the variables in $V$. The graph specifies the dependence structure of the variables and $P_V$ factorizes according to $G$ (see below). Given $G$ it is possible to ascertain if two sets of variables $A$ and $B$ are *conditionally independent* given another set of variables $D$, due to the global Markov property

$$A \perp\!\!\!\perp B \mid D, \text{ if } D \text{ separates } A \text{ from } B.$$

For a Markov network with a chordal graph $G$, the probability of a joint outcome $x$ factorizes as

$$P_V(x) = \frac{\prod_{c_i \in C} P_{c_i}(x_{c_i})}{\prod_{s_i \in S} P_{s_i}(x_{s_i})}.$$

Following this factorization the marginal likelihood of a dataset $\mathbf{X}$ given a Markov network with a chordal graph $G$ can be written

$$P(\mathbf{X}|G) = \frac{\prod_{c_i \in C} P_{c_i}(\mathbf{X}_{c_i})}{\prod_{s_i \in S} P_{s_i}(\mathbf{X}_{s_i})}.$$

By a suitable choice of prior distribution, the terms $P_{c_i}(\mathbf{X}_{c_i})$ and $P_{s_i}(\mathbf{X}_{s_i})$ can be calculated analytically. Let $a$ denote an arbitrary clique or separator containing the variables $X_a$ whose outcome space has the cardinality $k$. Further, let $n_a^{(j)}$ denote the number of occurrences where $X_a = x_a^{(j)}$ in

the dataset $\mathbf{X}_a$. Now assign the Dirichlet$(\alpha_{a_1}, \ldots, \alpha_{a_k})$ distribution as prior over the probabilities $P_a(X_a = x_a^{(j)}) = \theta_j$, determining the distribution $P_a(X_a)$. Now $P_a(\mathbf{X}_a)$ can be calculated as

$$P_a(\mathbf{X}_a) = \int_\Theta \prod_{j=1}^k (\theta_j)^{n_a^{(j)}} \cdot \pi_a(\theta) d\theta$$

where $\pi_a(\theta)$ is the density function of the Dirichlet prior distribution. By the standard properties of the Dirichlet integral, $P_a(\mathbf{X}_a)$ can be reduced to the form

$$P_a(\mathbf{X}_a) = \frac{\Gamma(\alpha)}{\Gamma(n_a + \alpha)} \prod_{j=1}^k \frac{\Gamma(n_a^{(j)} + \alpha_{a_j})}{\Gamma(\alpha_{a_j})}$$

where $\Gamma(\cdot)$ denotes the gamma function and

$$\alpha = \sum_{j=1}^k \alpha_{a_j} \qquad \text{and} \qquad n_a = \sum_{j=1}^k n_a^{(j)}.$$

When dealing with the marginal likelihood of a dataset it is most often necessary to use the logarithmic value $\log P(\mathbf{X}|G)$. Introducing the notations $v(c_i) = \log P_{c_i}(\mathbf{X}_{c_i})$ the logarithmic value of the marginal likelihood can be written

$$\log P(\mathbf{X}|G) = \sum_{c_i \in C} \log P_{c_i}(\mathbf{X}_{c_i}) - \sum_{s_i \in S} \log P_{s_i}(\mathbf{X}_{s_i}) = \sum_{c_i \in C} v(c_i) - \sum_{s_i \in S} v(s_i). \qquad (1)$$

The learning problem is to find a graph $G$ that optimizes the posterior distribution

$$P(G|\mathbf{X}) = \frac{P(\mathbf{X}|G)P(G)}{\sum_{G \in \mathcal{G}} P(\mathbf{X}|G)P(G)}.$$

Here $\mathcal{G}$ denotes the set of all graphs under consideration and $P(G)$ is the prior probability assigned to $G$. In the case where a uniform prior is used for the graphs the optimization problem reduces to finding the graph with the largest marginal likelihood.

## 3    Fundamental Properties and Characterization Results

In this section, we point out some properties of chordal graphs and clique graphs that can be utilized in the encodings of the learning problem. In particular, we develop a characterization of maximum weight spanning trees in terms of a *balancing condition* on separators.

The separators needed for determining the score (1) of a candidate Markov network are defined as follows. Given the cliques, we can form the *clique graph*, in which the nodes are the cliques and there is an edge between two nodes if the corresponding cliques have a non-empty intersection. We label each of the edges with this intersection and consider the cardinality of the label as its *weight*. The *separators* are the edge labels of a *maximum weight spanning tree* of the clique graph. Maximum weight spanning trees of arbitrary graphs can be found in polynomial time by reducing the problem to finding *minimum weight spanning trees*. This reduction consists of negating all the edge weights and then using any of the polynomial time algorithms for the latter problem [20]. There may be several maximum weight spanning trees, but they induce exactly the same separators, and they only differ in terms of which pairs of cliques induce the separators.

To restrict the search space we can observe that a chordal graph with $n$ nodes has at most $n$ maximal cliques [19]. This gives an immediate upper bound on the number of cliques chosen to build a Markov network, which can be encoded as a simple cardinality constraint.

### 3.1    Characterization of Maximum Weight Spanning Trees

To simplify the encoding of maximum weight spanning trees (and forests) of chordal clique graphs, we introduce the notion of *balanced spanning trees* (respectively, forests), and show that these two concepts coincide for chordal graphs. Then separators can be identified more effectively: rather than encoding an algorithm for finding maximum-weight spanning trees as constraints, it is sufficient to select a subset of the edges of the clique graph that is acyclic and satisfies the balancing condition expressible as a cardinality constraint over occurrences of nodes in cliques and separators.

**Definition 1 (Balancing)** *A spanning tree (or forest) of a clique graph is* balanced *if for every node $n$, the number of cliques containing $n$ is one higher than the number of labeled edges containing $n$.*

While in the following we state many results for spanning trees only, they can be straightforwardly generalized to spanning forests as well (in case the Markov networks are disconnected.)

**Lemma 2** *For any clique graph, all its balanced spanning trees have the same weight.*

*Proof:* This holds in general because the balancing condition requires exactly the same number of occurrences of any node in the separator edges for any balanced spanning tree, and the weight is defined as the sum of the occurrences of nodes in the edge labels. □

**Lemma 3 ([21, 22])** *Any maximum weight spanning tree of the clique graph is a junction tree, and hence satisfies the* running intersection property*: for every pair of nodes $c$ and $c'$, $(c \cap c') \subseteq c''$ for all nodes $c''$ on the unique path between $c$ and $c'$.*

**Lemma 4** *Let $T = \langle V, E_T \rangle$ be a maximum weight spanning tree of the clique graph $\langle V, E \rangle$ of a connected chordal graph. Then $T$ is balanced.*

*Proof:* We order the tree by choosing an arbitrary clique as the root and by assigning a depth to all nodes according to their distance from the root node. The rest of the proof proceeds by induction on the height of subtrees starting from the leaf nodes as the base case. The induction hypothesis says that all subtrees satisfy the balancing condition. The base cases are trivial: each leaf node (clique) trivially satisfies the balancing condition, as there are no separators to consider.

In the inductive cases, we have a clique $c$ at depth $d$, connected to one or more subtrees rooted at neighboring cliques $c_1, \ldots, c_k$ at depth $d + 1$, with the subtrees satisfying the balancing condition. We show that the tree consisting of the clique $c$, the labeled edges connecting $c$ respectively to cliques $c_1, \ldots, c_k$, and the subtrees rooted at $c_1, \ldots, c_k$, satisfies the balancing condition.

First note that by Lemma 3, any maximum weight spanning tree of the clique graph is a junction tree and hence satisfies the running intersection property, meaning that for any two cliques $c_1$ and $c_2$ in the tree, every clique on the unique path connecting them includes $c_1 \cap c_2$.

We have to show that the subtree rooted at $c$ is balanced, given that its subtrees are balanced. We show that the balancing condition is satisfied for each node separately. So let $n$ be one of the nodes in the original graph. Now each of the subtrees rooted at some $c_i$ has either 0 occurrences of $n$, or $k_i \leq 1$ occurrences in the cliques and $k_i - 1$ occurrences in the edge labels, because by the induction hypothesis the balancing condition is satisfied. Four cases arise:

1. The node $n$ does not occur in any of the subtrees.

   Now the balancing condition is trivially satisfied for the subtree rooted at $c$, because $n$ either does not occur in $c$, or it occurs in $c$ but does not occur in the label of any of the edges to the subtrees.

2. The node $n$ occurs in more than one subtree.

   Since any maximum weight spanning tree is a junction tree by Lemma 3, $n$ must occur also in $c$ and in the labels of the edges between $c$ and the cliques in which the subtrees with $n$ are rooted. Let $s_1, \ldots, s_j$ be the numbers of occurrences of $n$ in the edge labels in the subtrees with at least one occurrence of $n$, and $t_1, \ldots, t_j$ the numbers of occurrences of $n$ in the cliques in the same subtrees.

   By the induction hypothesis, these subtrees are balanced, and hence $t_i - s_i = 1$ for all $i \in \{1, \ldots, j\}$. The subtree rooted at $c$ now has $1 + \sum_{i=1}^{k} t_i$ occurrences of $n$ in the nodes (once in $c$ itself and then the subtrees) and $j + \sum_{i=1}^{j} s_i$ occurrences in the edge labels, where the $j$ occurrences are in the edges between $c$ and the $j$ subtrees.

We establish the balancing condition through a sequence of equalities. The first and the last expression are the two sides of the condition.

$$
\begin{aligned}
(1 + \sum_{i=1}^{j} t_i) - (j + \sum_{i=1}^{k} s_i) & \\
= 1 - j + \sum_{i=1}^{j}(t_i - s_i) & \quad \text{reordering the terms} \\
= 1 - j + j & \quad \text{since } t_i - s_i = 1 \text{ for every subtree} \\
= 1 &
\end{aligned}
$$

Hence also the subtree rooted at $c$ is balanced.

3. The node $n$ occurs in one subtree and in $c$.

   Let $i$ be the index of the subtree in which $n$ occurs. Since any maximum weight spanning tree is a junction tree by Lemma 3, $n$ must occur also in the clique $c_i$. Hence $n$ occurs in the label of the edge from $c_i$ to $c$. Since the subtree is balanced, the new graph obtained by adding the clique $c$ and the edge with a label containing $n$ is also balanced. Further, adding all the other subtrees that do not contain $n$ will not affect the balancing of $n$.

4. The node $n$ occurs in one subtree but not in $c$.

   Since there are $n$ occurrences of $n$ in any of the other subtrees, in $c$, or in the edge labels between $c$ and any of the subtrees, the balancing condition holds.

This completes the induction step and consequently, the whole spanning tree is balanced. □

**Lemma 5** *Assume $T = \langle V, E_B \rangle$ is a spanning tree of the clique graph $G_C = \langle V, E \rangle$ of a chordal graph that satisfies the balancing condition. Then $T$ is a maximum weight spanning tree of $G_C$.*

*Proof:* Let $T_M$ be one of the spanning trees of $G_C$ with the maximum weight $w$. By Lemma 4, this maximum weight spanning tree is balanced. By Lemma 2, $T$ has the same weight $w$ as $T_M$. Hence also $T$ is a maximum weight spanning tree of $G_C$. □

**Theorem 6** *For any clique graph of a chordal graph, any of its subgraphs is a maximum weight spanning tree if and only if it is a balanced acyclic subgraph.*

## 4  Representation as Constraints

In this section we first show how the structure learning problem of Markov networks is cast as a constraint satisfaction problem, and then formalize it concretely in the language of propositional logic, as directly supported by SMT solvers and easily translatable into conjunctive normal form as used by SAT and MAXSAT solvers. In ASP slightly different rule-based formulations are used.

The learning problem is formalized as follows. The goal is to find a *balanced* spanning tree (cf. Definition 1) for a set $C$ of cliques forming a Markov network and the set $S$ of separators induced by the tree structure. In addition, $C$ and $S$ are supposed to be optimal in the sense of (1), i.e., the overall *score* $v(C, S) = \sum_{c \in C} v(c) - \sum_{s \in S} v(s)$ is maximized. The individual score $v(c)$ for any set of nodes $c$ describes how well it reflects the interdependencies of the variables in $c$ in the data.

**Definition 7** *Let $V$ be a set of nodes representing random variables and $v : \mathbf{2}^V \to \mathbb{R}$ a scoring function. A* solution *to the Markov network learning problem is a set of* cliques $C = \{c_1, \dots, c_k\}$ *satisfying the following requirements viewed as abstract constraints:*

1. *Every node is included in at least one of the chosen cliques in $C$, i.e., $\bigcup_{i=1}^{k} c_i = V$.*

2. *Cliques in $C$ are maximal, i.e.,*

   (a) *for every $c, c' \in C$, if $c \subseteq c'$, then $c = c'$; and*
   (b) *for every $c \subseteq V$, if $edges(c) \subseteq \bigcup_{c' \in C} edges(c')$, then $c \subseteq c'$ for some $c' \in C$*

   *where $edges(c) = \{\{n, n'\} \subseteq c \mid n \neq n'\}$ is defined for each $c \subseteq V$.*

3. *The graph $\langle V, E \rangle$ with the set of edges $E = \bigcup_{c \in C} edges(c)$ is chordal.*

*4. The set $C$ has a balanced spanning tree labeled by a set of separators $S = \{s_1, \ldots, s_l\}$.*

*Moreover, the solution is* optimal *if it maximizes the overall score* $v(C, S)$.

The encodings of basic graph properties (conditions 1 and 2 above) are presented Section 4.1. The more complex properties (3 and 4) are addressed in Sections 4.2 and 4.3.

## 4.1 Graph Properties

We assume that clique candidates – which are the non-empty subsets of $V$ – are indexed from 1 to $2^{|V|}$. We often identify a clique with its index. Each clique candidate $c \subseteq V$ has an associated score $v(c)$. To encode the search space for Markov networks, we introduce, for every clique candidate $c$, a propositional variable $x_c$ denoting that $c$ is part of the learned network. We also introduce propositional variables $e_{n,m}$ that represent edges $\{n, m\}$ that are in at least one chosen clique.[1]

To formalize condition 1 of Definition 7, for every node $n$ we have the constraint

$$x_{c_1} \vee \cdots \vee x_{c_k} \tag{2}$$

where $c_1, \ldots, c_k$ are all cliques $c$ with $n \in c$.

To satisfy the maximality condition 2(a), we require that if a clique is chosen, then at least one edge in each of its super-cliques is not chosen. We first make the edges of the chosen cliques explicit by the next constraint for all $\{n, m\} \subseteq V$ and cliques $c_1, \ldots, c_k$ such that $\{n, m\} \subseteq c_i$.

$$e_{n,m} \leftrightarrow (x_{c_1} \vee \cdots \vee x_{c_k}) \tag{3}$$

Then for every clique candidate $c = \{n_1, \ldots, n_k\}$ and every node $n \in V \backslash c$ we have the constraint

$$x_c \rightarrow (\neg e_{n_1,n} \vee \cdots \vee \neg e_{n_k,n}) \tag{4}$$

where $e_{n_1,n}, \ldots, e_{n_k,n}$ represent all additional edges that would turn $c \cup \{n\}$ into a clique. For each pair of clique candidates $c$ and $c'$ such that $c \subset c'$, $\neg x_c \vee \neg x_{c'}$ is a logical consequence of the constraints (4). They are useful for strengthening the inferences made by SAT solvers.

For condition 2(b) we use propositional variables $z_c$ which mean that either $c$ or one of its super-cliques is chosen, and propositional variables $w_c$ which mean that all edges of $c$ are chosen. For 2-element cliques $c = \{n_1, n_2\}$ we have

$$w_c \leftrightarrow e_{n_1,n_2}. \tag{5}$$

For larger cliques $c$ we have

$$w_c \leftrightarrow w_{c_1} \wedge \cdots \wedge w_{c_k} \tag{6}$$

where $c_1, \ldots, c_k$ are all subcliques of $c$ with one less node than $c$. Hence $w_c$ is true iff all edges of $c$ are chosen. If all edges of a clique are chosen, then the clique itself or one of its super-cliques must be chosen. If $c_1, \ldots, c_k$ are all cliques that extend $c$ by one node, this is encoded as follows.

$$w_c \rightarrow z_c \tag{7}$$
$$z_c \leftrightarrow (x_c \vee z_{c_1} \vee \cdots \vee z_{c_k}) \tag{8}$$

## 4.2 Chordality

We use a straightforward encoding of the chordality condition (3) of Definition 7. The idea is to generate constraints corresponding to every $k \geq 4$ element subset $S = \{n_1, \ldots, n_k\}$ of $V$. Let us consider all cycles these nodes could form in the graph $\langle V, E \rangle$ of condition 3 in Definition 7. A cycle starts from a given node, goes through all other nodes, with (undirected) edges between two consecutive nodes, and ends in the starting node. The number of constraints can be reduced by two observations. First, the same cycle could be generated from different starting nodes, e.g., cycles $n_1, n_2, n_3, n_4, n_1$ and $n_2, n_3, n_4, n_1, n_2$ are the same. Second, generating the same cycle in two opposite directions, as in $n_1, n_2, n_3, n_4, n_1$ and $n_1, n_4, n_3, n_2, n_1$, is unnecessary. To avoid

redundant cycle constraints, we arbitrarily fix the starting node and require that the index of the second node in the cycle is lower than the index of the second last node. These restrictions guarantee that every cycle associated with $S$ is considered exactly once. Now, the chordality constraint says that if there is an edge between every pair of consecutive nodes in $n_1, \ldots, n_k, n_1$, then there also has to be an edge between at least one pair of two non-consecutive nodes. In the case $k = 4$, for instance, this leads to formulas of the form

$$e_{n_1,n_2} \wedge e_{n_2,n_3} \wedge e_{n_3,n_4} \wedge e_{n_4,n_1} \rightarrow e_{n_1,n_3} \vee e_{n_2,n_4}. \tag{9}$$

This encoding of chordality constraints is exponential in $|V|$ and therefore not scalable to large numbers of nodes. However, the datasets considered in Section 5 have only 6 or 8 variables, and in these cases the exponentiality is not an issue.

### 4.3  Separators

Separators for pairs $c$ and $c'$ of clique candidates can be formalized as propositional variables $s_{c,c'}$, meaning that $c \cap c'$ is a separator and there is an edge in the spanning tree between $c$ and $c'$ labeled by $c \cap c'$. The corresponding constraint is

$$s_{c,c'} \rightarrow x_c \wedge x_{c'}. \tag{10}$$

The lack of the converse implication formalizes the *choice* of the spanning tree, i.e., $s_{c,c'}$ can be false even if $x_c$ and $x_{c'}$ are true. The remaining constraints on separators fall into two cases.

First, we have cardinality constraints encoding the balancing condition (cf. Section 3.1): each variable occurs in the chosen cliques one more time than it occurs in the separators which label the spanning tree. Cardinality constraints are natively supported by some constraint solvers, or they can be reduced to Boolean constraints [23]. Second, the graph formed by the cliques with the separators as edges must be acyclic. We encode this through an inductive definition of trees: repeatedly remove *leaf nodes*, i.e., nodes with at most one neighbor, until all nodes have been removed. When applying this definition to a cyclic graph, some nodes will remain in the end. We define the *leaf level* for each node. A node is a *level 0 leaf* iff it has 0 or 1 neighbors in the graph. A node is a *level $l + 1$ leaf* iff all its neighbors except possibly one are level $j \leq l$ leaves. This definition is directly expressible as Boolean constraints. A graph with $m$ nodes is acyclic iff all its nodes are level $\lfloor \frac{m}{2} \rfloor$ leaves.

## 5  Experimental Evaluation

The constraints described in Section 4 can be alternatively expressed as MAXSAT, SMT, or ASP problems. We have used respective solvers in computing optimal Markov networks for datasets from the literature. The test runs were with an Intel Xeon E3-1230 CPU running at 3.20 GHz.

1. For the MAXSAT encodings, we tried out SAT4J (version 2.3.2) [24] and PWBO (version 2.2) [25]. The latter was run in its default configuration as well as in the UB configuration.

2. For SMT, we used the OPTIMATHSAT solver (version 5) [26].

3. For ASP, we used the CLASP (version 2.1.3) [27] and HCLASP (also v. 2.1.3) [28] solvers. The latter allows declaratively specifying search heuristics. We also tried the LP2NORMAL tool that reduces cardinality constraints to more basic constraints [29].

We consider two datasets, one containing risk factors in heart diseases and the other variables related to economical behavior [18], to be abbreviated by **heart** and **econ** in the sequel. For **heart**, the globally optimal network has been verified via (expensive) exhaustive enumeration. For **econ**, however, exhaustive enumeration is impractical due to the extremely large search space, and consequently the optimality of the Markov network found by stochastic search in [4] had been open until now. For both datasets, we computed the respective *score file* that specifies the score of each clique candidate, i.e., the log-value of its potential function, and the list of variables involved in that clique. The score files were then translated to be run with the different solvers. The MAXSAT and ASP solvers only support integer scores obtained by multiplying the original scores by 1000 and rounding. The SMT solver OptiMathSAT used the original floating point scores. The results are given in Table 1.

The **heart** data involves 6 variables giving rise to $2^6 = 64$ clique candidates in total and a search space of $2^{15}$ undirected networks of which a subset are decomposable. For instance, the ASP solver

|  | heart | econ | heart | econ |
|---|---|---|---|---|
| OPTIMATHSAT | 74 | - | 3930 kB | 139 MB |
| PWBO (default) | 158 | - | 3120 kB | 130 MB |
| PWBO (UB) | 63 | - | 3120 kB | 130 MB |
| SAT4J | 28 | - | 3120 kB | 130 MB |
| LP2NORMAL+CLASP | 111 | - | 8120 kB | 1060 MB |
| CLASP | 5.6 | - | 197 kB | 4.2 MB |
| HCLASP | 1.6 | $310 \times 10^3$ | 203 kB | 4.2 MB |

Table 1: Summary of results: Runtimes in seconds and sizes of solver input files

HCLASP traversed a considerably smaller search space that consisted of 26651 (partial) networks. This illustrates the power of branch-and-bound type algorithms behind the solvers and their ability to prune the search space. On the other hand, the **econ** dataset is based on 8 variables giving rise to a much larger search space $2^{28}$. We were able to solve this instance optimally with one solver only, HCLASP, which allows for a more refined control of the search heuristic: we forced HCLASP to try cliques in an ascending order by size, with greatest cliques first. This allowed us to find the global optimum in about 14 hours, after which 3 days is spent on the proof of optimality.

## 6 Conclusions

Boolean constraint methods appear not to have been earlier applied to learning of undirected Markov networks. We have introduced a generic approach in which the learning problem is expressed in terms of constraints on variables that determine the structure of the learned network. The related problem of structure learning of Bayesian networks has been addressed by general-purpose combinatorial search methods, including MAXSAT [30] and a constraint-programming solver with a linear-programming solver as a subprocedure [31, 32]. We introduced explicit translations of the generic constraints to the languages of MAXSAT, SMT and ASP, and demonstrated their use through existing solver technology. Our method thus opens up a novel venue of research to further develop and optimize the use of such technology for network learning. A wide variety of possibilities does exist also for using these methods in combination with stochastic or heuristic search.

## Footnotes

[1]As the edges are undirected, we limit to $e_{n,m}$ such that the ordering of $n$ and $m$ according to some fixed ordering is increasing, i.e., $n < m$. Under this assumption, $e_{m,n}$ for $n < m$ denotes $e_{n,m}$.

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
