[Reviews · NeurIPS 2013]

Submitted by Assigned_Reviewer_5

The authors recast learning of chordal Markov networks as a constraint satisfaction problem.

The paper is generally well-written and the authors show that their method is able to compute the best structure for some problems for which this was previously unknown.

Having said that, the method will obviously not scale to examples much larger than those considered (owing to the exponential growth in the number of propositional variables required).

It would also be useful to make some comparison to the literature on learning DAGs (e.g. papers by Koivisto and Sood) given that obviously chordal Markov networks are a subclass of DAGs.

Summary: A well-written paper that advances the state of the art in exact structure learning.
A bit of additional comparison to work on learning Bayesian Networks would be useful.

Submitted by Assigned_Reviewer_7

The authors formulate the problem of learning an undirected graphical model in terms of a large (intractable) constraint satisfaction problem (CSP)In particular, given observations of the variables, X, they focus on finding the junction tree, T, that with the largest marginal likelihood p(X|T). Rather than blindly searching in the space of all junction trees, they formulate a weighted constraint satisfaction problem and propose to use off-the-shelf CSP solver to find the best junction tree.

The majority of the paper is on formulating the CSP, including characterizing a junction tree as a collection of cliques and separators over a balanced max weight spanning tree. While this paper is very well written and appears to be novel it may be of little practical value simply because formulating the CSP problem requires an exponential number of constraints - in other words, what do we gain by casting the structure learning problem as a CSP? I would liked to have seen an approximate CSP that relaxes some conditions but remains tractable, or a bounded form of the CSP that learns the most likely junction tree with bounded clique size. Last, I'm not sure how a non-uniform prior on graph structures could be incorporated, but we would prefer a means to learn sparse/low treewidth models.
Summary: Authors propose a novel formulation of structure leaning as a CSP and introduce a nice tool - the balanced spanning tree - for analyses and algorithms involving junction trees. However, the approach is of little practical value because the CSP requires an exponential number of constraints.
Author Feedback

Author rebuttal: REVIEWER 5:
Yes, clarifying that we assume chordality is useful, and will revise the title, abstract and elsewhere to emphasize this assumption.

REVIEWER 6:
The reviewer's summary of the proof of Lemma 4 about the balancing condition is accurate. We may have been a bit pedantic in spelling out the details of the proof, but on the other hand, simply saying that the balancing condition "obviously" holds because of the running intersection property would not be very informative either, and we would rather err on the side of giving too much details rather than too little.

The standard Bayesian approach we use for model learning is statistically consistent for choosing the correct dimensionality, since prior distribution assigned to model parameters acts as a regularizer. This property is so widely established in the literature that we did not consider it to be necessary to emphasize the aspect in the paper. However, in the revision we will clarify the matter.

About the assumption concerning chordality. Machine learning literature is nearly exclusively restricted to learning of the chordal, i.e. decomposable Markov networks, due to the possibility to perform Bayesian or maximum likelihood inference in closed form. Since SAT-based approaches have not been previously developed for structural learning of any kind of Markov networks to the best of our knowledge, we find it appropriately motivated to develop first a solution under the chordality assumption. This can likely spawn ways to handle the non-chordal graphs in the future.

REVIEWER 7:
We find it unreasonable to criticize our work on the grounds that CSP problem requires an exponential number of constraints. Notice that this exponentiality is in the number of variables, and the encoding is polynomial in the number of candidate cliques of the Markov network, and this is no different from earlier alternative (i.e. other than SAT/constraints) methods.
As also concluded by the other reviewers, we present how the model learning can be exactly translated to a SAT framework, which represents an important first step towards developing scalable methods for larger networks. According to our opinion it is more fruitful to move towards approximate constraints once it is thoroughly understood how exact approaches can be built under this framework.